# BanglaAbuseMeme: A Dataset for Bengali Abusive Meme Classification

**Mithun Das**
IIT Kharagpur
West Bengal, India
mithundas@iitkgp.ac.in

**Animesh Mukherjee**
IIT Kharagpur
West Bengal, India
animeshm@cse.iitkgp.ac.in

## Abstract

The dramatic increase in the use of social media platforms for information sharing has also fueled a steep growth in online abuse. A simple yet effective way of abusing individuals or communities is by creating memes, which often integrate an image with a short piece of text layered on top of it. Such harmful elements are in rampant use and are a threat to online safety. Hence it is necessary to develop efficient models to detect and flag abusive memes. The problem becomes more challenging in a low-resource setting (e.g., Bengali memes, i.e., images with Bengali text embedded on it) because of the absence of benchmark datasets on which AI models could be trained. In this paper we bridge this gap by building a Bengali meme dataset. To setup an effective benchmark we implement several baseline models for classifying abusive memes using this dataset[1]. We observe that multimodal models that use both textual and visual information outperform unimodal models. Our best-performing model achieves a macro F1 score of 70.51. Finally, we perform a qualitative error analysis of the misclassified memes of the best-performing text-based, image-based and multimodal models.

*Disclaimer: This paper contains elements that one might find offensive which cannot be avoided due to the nature of the work.*

## 1 Introduction

In recent times, *Internet memes* have become commonplace across social media platforms. A meme is an idea usually composed of an image and a short piece of text on top of it, entrenched as part of the image (Pramanick et al., 2021b). Memes are typically jokes, but in the current Internet culture they can be far beyond jokes. People make memes as they are free of cost but can impress others and help to accrue social capital. Bad actors however use memes to threaten and abuse individuals or specific target communities. Such memes are collectively known as abusive memes on social media. Owing to their naturally viral nature, such abusive memes can ignite social tensions, tarnish the reputation of the platforms that host them (Statt, 2017), and may severely affect the victims psychologically (Vedeler et al., 2019). Therefore, controlling the spread of such abusive memes is necessary and the first step toward this is to efficiently detect them.

In the past few years a number of studies have tried to take initiative to detect and control the effect of abusive memes on different social media platforms. However most of these are concentrated around memes that have English as the text component (Sabat et al., 2019; Gomez et al., 2020; Suryawanshi et al., 2020; Pramanick et al., 2021a). Further, several multimodal vision language models have been explored, but again they are limited to English. Efforts in other languages is low and this is particularly true for resource-impoverished languages like Bengali (aka *Bangla*). In this paper we consider the problem of detecting Bangla abusive memes (see Figure 1 for examples) across social media platforms. The motivation comes from the fact that Bangla is the seventh most spoken language (ber) in the world having over 210 million speakers, with around 100 million Bengali speakers in Bangladesh and about 85 million speakers in India. It is the official language of Bangladesh and one of the officially recognized languages in the constitution of India. Besides Bangladesh and India, Bengali is spoken in many other countries, including the United Kingdom, the United States, and several countries in the Middle East (Britannica, 2022). The other important reason for considering Bangla is linked to the several smearing incidents in Bangladesh and India, such as slandering moves against famous political leaders, celebrities, and social media personalities, online anti-religious propaganda, and cyber harassment (Das et al., 2022) that have inflicted the online world.

---

[1]We make our code and dataset public for others on https://github.com/hate-alert/BanglaAbuseMeme

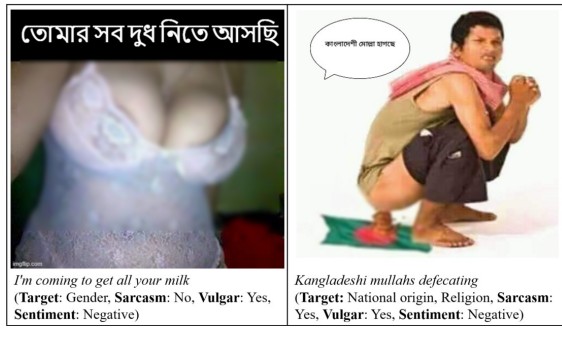

*I'm coming to get all your milk*
(**Target**: Gender, **Sarcasm**: No, **Vulgar**: Yes, **Sentiment**: Negative)

*Kangladeshi mullahs defecating*
(**Target**: National origin, Religion, **Sarcasm**: Yes, **Vulgar**: Yes, **Sentiment**: Negative)

Figure 1: Examples of abusive memes.

Our key contributions in this paper are as follows.

- To bridge the gap in Bangla abusive meme research we release a dataset – **BanglaAbuse-Meme** – to facilitate the automatic detection of such memes. The dataset comprises gold annotations for 4,043 Bengali memes, among which 1,515 memes are abusive while the rest are non-abusive. Further, each instance has been marked with the following labels (a) whether vulgar or not, (b) whether sarcastic or not, (c) the sentiment in the meme (positive, negative and neutral, and (d) the target community being attacked. We believe that our comprehensive annotations will enable us to gain a deeper understanding of the dynamics of Bengali memes as well as help future research.
- We implement several baseline models to identify abusive memes automatically. The model variants include purely text-based, purely image-based, and fusion-based multimodal approaches. We observe that our multimodal model **CLIP(L)** outperforms the other variants achieving an overall macro F1-score of 70.51. We also observe that the multimodal model performs well for most of the target communities, while this is not true for the text/image-based baselines.
- We perform qualitative error analysis of a sample of memes of the best text, image and multimodal models where the models misclassify some of the test instances. While text-based approaches fail in the absence of toxic words, image-based approaches fail when the image is typically out-of-context. The multimodal model fails when the abuse is implicit in nature.

## 2 Related work

**Abusive meme datasets**: In an effort to develop resources for multimodal abusive meme detection, several datasets have been constructed. Sabat et al. (2019) created a dataset of 5,020 memes for hate meme detection by crawling Google Images. Gomez et al. (2020) contributed another multimodal dataset (MMHS150K) for hate speech detection which has 150K posts collected from Twitter. Similarly, Chandra et al. (2021) developed a dataset for detecting antisemitism in multimodal memes by crawling datasets from both Twitter and Gab. Suryawanshi et al. (2020) created another dataset of 743 memes annotated as offensive or not-offensive. The dataset has been built by leveraging the memes related to the 2016 US presidential election. Pramanick et al. (2021a) also built another dataset to detect harmful memes containing around 3.5K memes related to COVID-19. In addition, to boost the research around multimodal abusive memes, several shared tasks have been organized. Facebook AI (Kiela et al., 2020) introduced another dataset of 10K+ posts comprising labeled hateful and non-hateful as part of the Hateful Memes Challenge. There are two previous works on Bengali meme detection. The first work, conducted by Karim et al. (2022), involved extending the Bengali hate speech dataset (Karim et al., 2020) by labeling 4,500 memes. Hossain et al. (2022) developed a dataset of 4,158 memes with Bengali and code-mixed captions. In the former work the authors have not made the data public (although they put a github link in the paper albeit without the link to the dataset). Further the data collection and annotation process is not discussed in detail, the target communities are not labeled and the inter-annotator agreement is also not reported. The latter work does not make the data public. Here again the target labels are missing making it difficult to ascertain how diverse the data is.

In our work, we aimed to overcome the drawbacks of these previous studies by not only labeling memes as abusive or non-abusive but also further annotating them with labels such as vulgar, sarcasm, sentiment, and the targeted community. This richness of our data provides a more holistic understanding of the dynamics of Bengali memes.

**Multimodal abusive meme detection**: With regards to abusive meme detection, several techniques based on diverse model architectures have been investigated. Sabat et al. (2019) used a

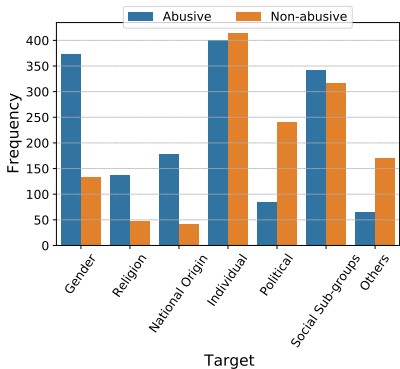

Figure 2: Distribution of data points across the target communities.

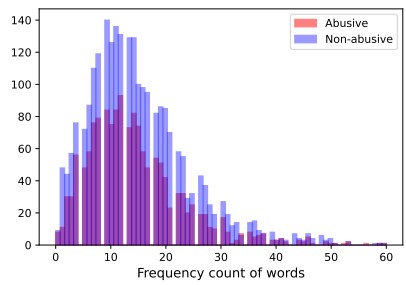

Figure 3: Histogram of the word count of the meme text for each class.

plethora of models, including BERT, VGG16, and MLP. Gomez et al. (2020) explored several models such as FCM, Inception-V3, LSTM, etc. With the advancement of the vision language models, several approaches, such as VisualBERT (Li et al., 2020), ViLBERT (Lu et al., 2019), MMBT (Kiela et al., 2019), UNITER (Chen et al., 2020), and others (Velioglu and Rose, 2020; Pramanick et al., 2021b; Chandra et al., 2021) have been explored. Recent studies have also attempted several data augmentation techniques (Velioglu and Rose, 2020; Lee et al., 2021) to improve classification performance. While previous methods were proposed for abusive speech detection for the English language, we focus on detecting Bengali abusive memes.

## 3 Dataset creation

Here we discuss the data collection strategy, annotation guidelines, and the statistics of the annotated dataset.

**Data collection and sampling**: In order to construct the abusive meme dataset, we first build a lexicon of 69 offensive Bengali terms. This lexicon comprises words that target individuals or different protected communities. In addition, we include the targeted community's name in the lexicon. The choice is made to extract random hateful/offensive memes about a community that do not contain abusive words. Then, using these lexicons, similar to Pramanick et al. (2021a) we perform a keyword-based web search on various platforms such as Google Image, Bing, etc. We also scrape various pages and groups from Facebook and Instagram. To download the images, we used an extension[2] of Google Chrome. Unlike the Hateful Memes

challenge (Kiela et al., 2020), which contributed synthetically generated memes, our dataset consists of memes curated from the real-world. Once the images are downloaded, we apply the following data sampling techniques before moving forward with the actual annotation. We remove the memes having no text or having text in different languages other than Bengali. We also remove memes having very low resolution where the meme text is unreadable.

**Annotation strategy**: To identify whether a meme is abusive or non-abusive, we hire five undergraduate students for our annotation task: three of them were males, and the other two were females, and they are all in the age range of 24 to 29 years. All the undergraduate students are native Bengali speakers and have been recruited on a voluntary basis. We paid them fairly for their work as per the standard local compensation rate[3]. The annotation process was supervised by an expert Ph.D. student with over four years of experience dealing with malicious social media content. Each meme in our dataset contains five kinds of annotations. First, whether the meme is abusive or not. Second, the target communities of the meme, if any. The targets in our dataset belong to one of the following seven categories – *gender*, *religion*, *national origin*, *individual*, *political*, *social sub-groups*, and *others*. Third, whether the meme is vulgar or not. Fourth, whether the meme is sarcastic or not. Last, the sentiment labels (positive/neutral/negative) associated with the meme. For non-abusive memes, if the target is absent, no target label is assigned to that meme.

*Annotation codebook*: In order to ensure accurate and consistent annotation, detailed annotation guidelines are essential for annotators to determine suitable labels for each meme. To achieve this, we

---

[2]https://download-all-images.mobilefirst.me/

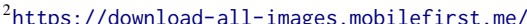

[3]We gave them one Indian rupee for annotating each meme.

| Abusive | # | Non-abusive | # |
|---|---|---|---|
| না (No) | 201 | না (No) | 379 |
| করে (By doing) | 188 | করে (By doing) | 310 |
| আর (And) | 180 | আর (And) | 251 |
| বাঙাল (Bangal) | 108 | আমি (I) | 182 |
| আমি (I) | 88 | তো (So) | 152 |
| ঘটি (Ghoti) | 86 | আমার (My) | 140 |
| তো (So) | 85 | এই (This) | 137 |
| কি (What) | 84 | কি (What) | 127 |
| হয় (Is) | 83 | হয়ে (Become) | 120 |
| হিন্দু (Hindu) | 82 | থেকে (From) | 116 |

Table 1: Top-10 most frequent words per class.

have developed a comprehensive annotation codebook that provides clear instructions and examples, enabling all participants to understand the labeling task effectively. Our codebook incorporates annotation guidelines and numerous illustrative examples tailored explicitly for identifying abusive memes. We shared the annotation codebook with the annotators and performed multiple training rounds. Through these training sessions, we familiarized the annotators with the guidelines and equipped them with the necessary knowledge to determine appropriate labels for the given memes (see Appendix A for more details).

**Annotation procedure**: Prior to the start of the actual annotation, we require a pilot gold-label dataset to guide the annotators, as mentioned above. Initially, we annotated 100 memes, out of which 60 were abusive and 40 were labeled as non-abusive. We take ten samples from each label and incorporate them into the annotation codebook, and the rest data points are used for evaluating the trial annotation discussed below.

*Trial annotation*: During the trial annotation task, we gave the annotators 80 memes and asked them to label the memes according to the annotation guidelines. We instructed the annotators to keep the annotation codebook open while doing the annotation to have better clarity about the labeling scheme. After the annotators finished this set, we consulted with them regarding their incorrect annotations. The trial annotation is an important stage for any dataset creation process as these activities help the annotators better understand the task by correcting their mistakes. In addition, we collected feedback from annotators to enrich the main annotation task.

*Main annotation*: After the trial stage, we proceeded with the main annotation task. We use the open-source data labeling tool *Label Studio*[4] for

[4] https://labelstud.io/

this task, which is deployed on a Heroku instance. We provided a secure account to each annotator where they could annotate and track their progress.

Based on the guidelines provided, three independent annotators have annotated each meme, and then majority voting was applied to determine the final label. Initially, we provided a small batches of 100 memes for annotation and later expanded it to 500 memes as the annotators became more efficient. After completing each batch of annotations, we discussed the errors they made in the previous batch to preserve the annotators' agreement. Since abusive memes can be highly polarizing and adverse, the annotators were given plenty of time to finish the annotations. To choose the target community of a meme, we rely on the majority voting of the annotators. Exposure to online abuse could usher in unhealthy mental health issues (Ybarra et al., 2006; Guardian, 2017). Hence, the annotators were advised to take frequent breaks and not do the annotations in one sitting. Besides, we also had weekly meetings with them to ensure that the annotations did not affect their mental health.

**Final dataset**: Our final dataset consists of 4,043 Bengali memes, out of which 1,515 have been labeled as abusive and the remaining 2,528 as non-abusive. Next, 1,664 memes are labeled as sarcastic, while the remaining 2,379 are labeled as not sarcastic. Further, 1,171 memes are labeled as vulgar, while 2,872 memes are labeled as not vulgar. Finally, 592 memes are labeled as having a positive sentiment, 1,414 memes as neutral, and 2,037 memes as having a negative sentiment. We achieved an inter-annotator agreement of 0.799, 0.801, 0.67, and 0.72 for the abusive, vulgar, sarcasm, and sentiment labeling tasks, respectively, using the Fleiss' $\kappa$ score. These scores are better than the agreement scores on other related hate/abusive speech tasks (Ousidhoum et al., 2019; Guest et al., 2021). We show the target distributions of the dataset in Figure 2. In Figure 4, we show the overlap between the (non)-abusive vs. (non)-sarcastic. We observe that a meme an abusive meme might not be necessarily sarcastic. Figure 5 shows that most abusive posts are also vulgar. In Figure 6, we show the overlap between the abusive vs. sentiment and non-abusive vs. sentiment classes. It indicates that abusive memes mostly have negative sentiments, whereas non-abusive memes are mostly positive or neutral.

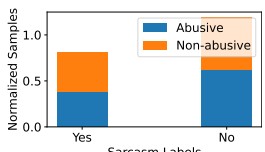

Figure 4: Abusive X Sarcasm.

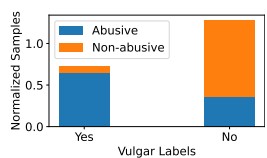

Figure 5: Abusive X Vulgar.

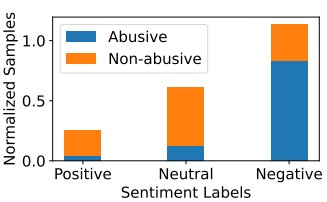

Figure 6: Abusive X Sentiment.

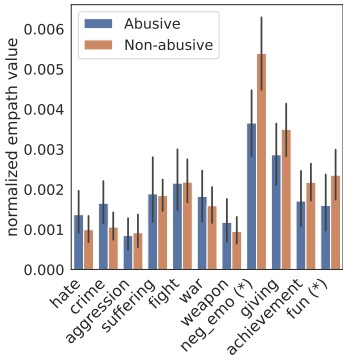

Figure 7: Lexical analysis of the meme texts using Empath. We report the mean values for several categories of Empath. * indicates ($p < 0.01$) using the M-W U test (McKnight and Najab, 2010).

| M | Model | Acc | M-F1 | F1(A) | P(A) | R(A) |
|---|-------|-----|------|-------|------|------|
| T | m-BERT[†] | 64.50 | 61.53 | 51.13 | 53.05 | 50.15 |
| | MuRIL[†] | 66.41 | 65.18 | 58.84 | 54.44 | 64.43 |
| | BanglaBERT[†] | 65.79 | 63.90 | 55.81 | 54.30 | 57.74 |
| | XLMR[†] | 67.79 | 65.73 | 57.63 | 57.09 | 59.03 |
| I | VGG16[†] | 67.47 | 64.77 | 55.21 | 57.54 | 53.80 |
| | ResNet-152[†] | 67.82 | 64.94 | 54.99 | 57.98 | 52.68 |
| | VIT[†] | 69.37 | 67.72 | 60.50 | 58.73 | 62.71 |
| | VAN | 66.33 | 64.64 | 57.31 | 55.24 | 60.21 |
| T + I | MU+RN(C)[†] | 67.30 | 65.34 | 57.25 | 56.49 | 58.42 |
| | XLM+RN(C)[†] | 69.30 | 67.13 | 58.75 | 59.19 | 58.48 |
| | MU+VIT(C)[†] | 69.62 | 67.15 | 58.29 | 60.40 | 56.95 |
| | XLM+VIT(C) | 69.57 | 68.19 | 61.62 | 58.49 | 65.28 |
| | CLIP(C)[†] | **72.02** | 69.92 | 62.01 | **63.20** | 60.98 |
| | MU+RN(L)[†] | 67.22 | 64.72 | 55.36 | 56.73 | 54.19 |
| | XLM+RN(L)[†] | 69.27 | 66.77 | 57.74 | 59.98 | 56.05 |
| | MU+VIT(L)[†] | 68.19 | 66.13 | 57.82 | 57.47 | 58.28 |
| | XLM+VIT(L)[†] | 68.95 | 66.26 | 56.80 | 59.44 | 54.73 |
| | CLIP(L)[*] | 71.78 | **70.51** | **64.60** | 61.44 | 68.70 |

Table 2: Performance comparisons across the different models. A: abusive class, P: precision, R: recall. The best performance in each column is marked in **bold**, and the second best is underlined. The best performing **CLIP(L)** model denoted by ⋆ is significantly different compared to other models marked by † as per the M-W U test ($p < 0.05$).

## 3.1 Text analysis

We perform the following analysis obtained from the meme text.

**Distribution of word length**: Figure 3 shows the average length of the extracted meme texts in terms of the number of words per class. We plot the histogram of word counts and find that both abusive and non-abusive meme texts follow a similar distribution. Further, we notice that most meme texts have lengths between 10 and 15 words.

**Most frequent words**: Table 1 shows the top 10 most frequent words in the meme texts of the annotated dataset for each class. Before counting the most frequent words, we use the BNLP (Sarker, 2021) library to remove the stop words. We observe that the target communities' names, such as Ghoti, Bangal, Hindu, etc., are more prevalent in the abusive class. We also create word clouds and find the presence of abusive words like *'Malaun'* (hateful word targeting 'Bengali Hindus'), *'Kangladeshi'* (hateful word targeting 'Bangladeshi'), *'Rendi'* (bitch), etc., in the abusive class. The presence of such words is relatively less in the non-abusive class.

**Empath analysis**: In order to understand the dataset better, we further identify important lexical categories present in the OCR extracted meme texts[5] using Empath (Fast et al., 2016), which has 189 such pre-built categories. First, we select categories excluding the topics irrelevant to abusive speech, e.g., technology and entertainment. We report the relevant categories in Figure 7. Abusive meme texts scored high in categories like 'hate', 'crime', 'suffering', 'fight', 'war' and 'weapon'. Non-abusive meme texts score higher on topics such as 'negative emotion (neg_emo)', 'giving', 'achievement' and 'fun'.

## 3.2 Image analysis

One of the important component in any meme is the presence/absence of a facial image. Hence, we attempt to analyze faces that are present in the meme. For this purpose, we use the FairFace library (Karkkainen and Joo, 2021). For a given image, first, we check if a face is present or not. If present, we further study the gender and age associated with the face. We observe that 33.2% of memes have no faces recognized, out of which

---

[5]Translated to English using Google translator.

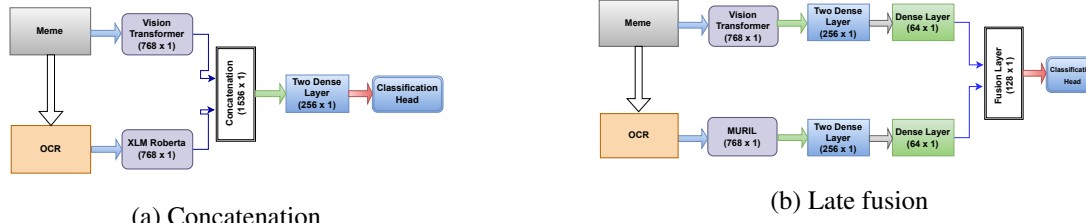

(a) Concatenation         (b) Late fusion

Figure 8: Fusion based models.

| Target | Acc | M-F1 | F1(A) | P(A) | R(A) |
|---|---|---|---|---|---|
| Gender | 40.11 | 40.09 | 41.39 | 74.30 | 28.68 |
| Religion | 61.62 | 54.31 | **72.58** | 77.68 | **68.11** |
| National Origin | 50.90 | 47.50 | 60.86 | **86.59** | 46.92 |
| Individual | 57.91 | 52.81 | 37.29 | 69.86 | 25.43 |
| Political | 50.00 | 48.42 | 39.40 | 28.80 | 62.35 |
| Social Sub-groups | 54.69 | 51.97 | 40.55 | 63.74 | 29.73 |
| Others | **78.90** | **71.46** | 56.89 | 66.00 | 50.00 |

Table 3: Zero-shot target-wise performance.

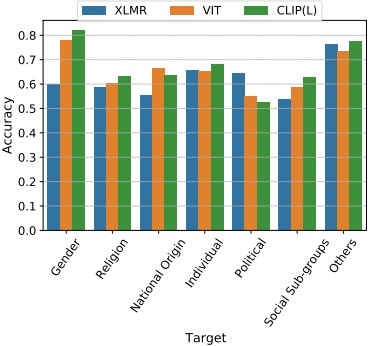

Figure 9: Target-wise performance. Only the best-performing models for each modality are shown.

13.3% are from the abusive class. Further, we observe that the mean number of faces for the abusive and non-abusive classes are 2.17 and 2.43, respectively. In the abusive class, 58.45% of faces are male, 41.54% are female, and around 68.24% of the female faces are between the 20-39 age group. For the non-abusive class, 73.20% of faces are male, and 26.79% are female; the female age group 20-39 makes up around 73.87% of the faces in this class. The key observation is that the % of females in the abusive class is double that in the non-abusive class, which possibly indicates that girls/women are extremely frequent victims of abuse.

## 4 Methodology

This section presents a suite of models we have used for Bengali abusive meme detection ranging from text-based, image-based, and, finally, fusion-based multimodal models. The techniques are discussed below.

**Text-based classification**: Here, the idea is to classify the memes solely based on the embedded text presented as part of the image. We use Easy-OCR[6] to extract the texts of a meme and apply standard pre-processing techniques on the post to remove URLs, mentions, and special characters. We experiment with the following transformer-based models for the classification, **m-BERT** (Devlin et al., 2019), **MurIL** (Khanuja et al., 2021), **XLM-Roberta** (Conneau et al., 2020) and **BanglaBERT** (Bhattacharjee et al., 2022).

**Image-based classification**: Here we pose the problem as an image classification task and experiment with the following pre-trained models: **VGG16** (Simonyan and Zisserman, 2014), **ResNet-152** (He et al., 2016), **Vision transformer(VIT)** (Dosovitskiy et al., 2021) and **Visual Attention Network(VAN)** (Guo et al., 2022).

**Multi-modal classification**: The unimodal models we discussed so far cannot leverage the relationship between the text and image in the meme. Thus, we attempt to combine both modalities meaningfully to capture the benefits of textual and visual features. We experiment with two different techniques of fusion - (1) *Concatenation* (C) and (2) *Late fusion* (L). For concatenation, the pre-trained features from both text-based and image-based models are concatenated and then passed through the MLP for the classification (see Figure 8 (a)). In the case of late fusion, the extracted text and image features are fed through MLP first, and then the intermediate layers' features are concatenated, and a final MLP is used for classification (see Figure 8 (b)). We choose the top two best-performing text and image-based models and construct the following fusion variants – (i) **MurIL + VIT**, (ii) **MurIL + ResNet-152**, (iii) **XLMR + VIT**, (iv) **XLMR + ResNet-152**. We also used **CLIP** (Radford et al., 2021), a pre-trained visual-linguistic model to cap-

---

[6]https://github.com/JaidedAI/EasyOCR

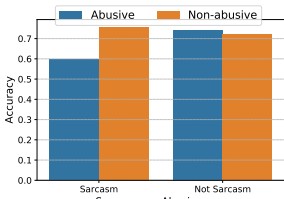
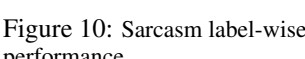

Figure 10: Sarcasm label-wise performance.

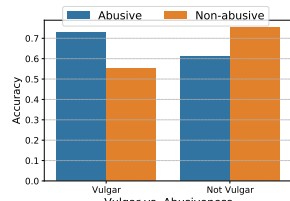

Figure 11: Vulgar label-wise performance.

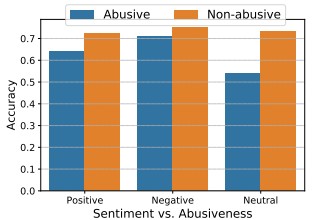

Figure 12: Sentiment label-wise performance.

ture the overall semantics of the memes due to its excellent performance in several harmful meme classification tasks (Maity et al., 2022; Kumar and Nanadakumar, 2022).

| Train | Test | Acc | M-F1 | F1(A) | P(A) | R(A) |
|-------|------|-----|------|-------|------|------|
| Hindi | Hindi | 71.64 | 71.46 | 73.77 | 75.20 | 72.40 |
| Hindi | Bengali | 52.28 | 52.28 | 51.87 | 41.85 | 68.19 |
| Bengali | Bengali | 70.08 | 69.28 | 64.30 | 58.44 | 71.47 |
| Bengali | Hindi | 50.46 | 45.07 | 27.86 | 70.44 | 17.36 |

Table 4: Comparison with existing dataset in another language (Maity et al., 2022).

## 4.1 Experimental setup

We evaluate our models using $k$-fold stratified cross-validation. For all the experiments, we set $k$ to 5 here, and for each fold, we use 70% data for training, 10% for validation, and the rest 20% for testing. We use the same test set across all the models to ensure a fair comparison. For all the **unimodal** neural network models, the internal layer has two fully connected dense layers of 256 nodes, reduced to a feature vector of length 2 (abusive vs non-abusive). We have two variants for fusion-based techniques. For **concatenation**, we first concatenate the text and image embeddings and pass the concatenated features through two fully connected dense layers of 256 nodes, which further maps to a feature vector of length 2. For the **late fusion**, the unimodal text and visual features are separately passed through two dense layers of size 256, which are finally fed to another dense layer of 64 nodes. Finally, we concatenate all the nodes and reduce them to a feature vector of length 2. In addition, we conduct an out-of-target experiment to evaluate how our model performs for an unseen target community. We exclude one target community from our training and validation data and build our model using the remaining target communities. We call this the zero-shot setting for the excluded target. We then assess the model's performance on that excluded target community. This experiment is conducted using our best classification model. For this experiment, we use 85% of the data for training and 15% for validation (see Appendix C for more details).

## 5 Experimental results

### 5.1 Abusive meme detection

Table 2 shows the performance for all the models. From the table, we observe the following. Among the *text-based models*, XLMR performs the best with an accuracy of 67.79% and a macro-F1 score of 65.73%. The MurIL model performs the second best in terms of macro-F1(65.18%) score. For *image-based models*, the VIT model performs the best (acc: 69.37%, macro-F1: 67.72%) and the ReseNet-152 model performs second best (acc: 67.82%, macro-F1: 64.94%). In the *multimodal* setup, we notice CLIP(L) exhibits the best performance in terms of macro-F1 score (70.51%) closely followed by CLIP(C) (macro-F1: 69.92%) at the second best position[7].

### 5.2 Target-wise performance

Figure 9 shows the target-wise performance of the best unimodal and multimodal models. We observe that the performance of the unimodal models varies across all target communities, while the multimodal model consistently reports good performance across all target communities except the political target community.

**Zero-shot**: In Table 3, we present the performance of zero-shot target-wise classification for the best multimodal CLIP(L) model. Except for the "Others" category, all other targets exhibit macro F1-

---

[7]We also investigate the performance of a multi-task model where abusive meme detection is the main task and vulgarity detection, sentiment classification and sarcasm detection are used as auxiliary tasks. Please refer to Appendix E for the results obtained from this model.

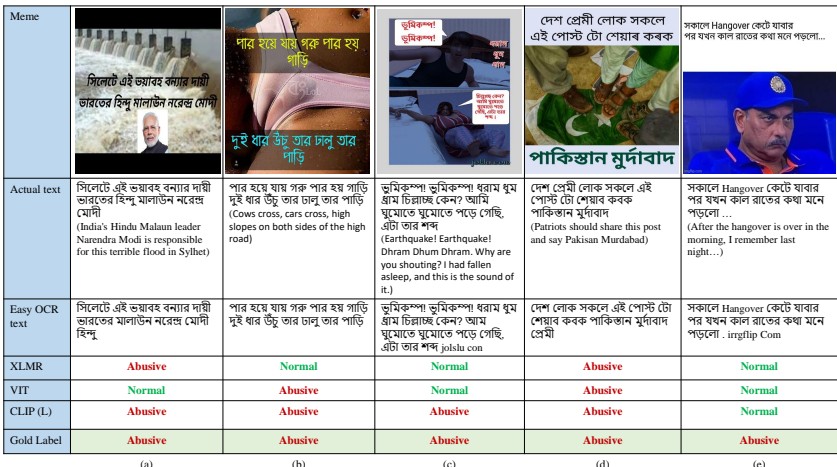

Figure 13: Inference results on the examples from the test set for the best text-based, vision-based, and multimodal models.

scores around or below 50%. While the precision and F1-score for the abusive class is high for the religion target, the increased misclassification in the non-abusive class adversely affects the overall macro F1-score. This finding underscores the need for more generalizability in abusive meme detection schemes across unknown target communities. In other words, developing effective strategies that can enhance performance in unseen target communities and improve detection capabilities is crucial.

## 5.3 Performance across different dimensions

In addition, we present the performance of the best multimodal model for the memes in the following buckets (i) sarcastic vs non-sarcastic (Figure 10) (ii) vulgar vs non-vulgar (Figure 11), (iii) positive vs negative vs neutral sentiments (Figure 12). We observe that the sarcastic memes in the abusive category exhibit poor performance compared to other categories. Further, the abusive memes in the vulgar subset shows the best performance while in the non-vulgar subset it is inferior. Finally, we observe that in all sentiment subsets the performance for non-abusive memes is similar. On the other hand, in the negative sentiment subset abusive memes are more accurately detected compared to the other two subsets.

## 5.4 Additional experiment

In addition, we performed another experiment to understand the performance of model transferability of abusive meme detection with another language dataset. For this, we use the cyberbullying detection dataset in Hindi (code-mixed) consisting of 5,854 memes created by Maity et al. (2022). Our experimentation focused exclusively on the CLIP(L) model, which exhibited superior performance among all the models considered. In this experimental setup, we trained the model using the Bengali language and evaluated its performance using the Hindi language and vice versa. We outline the results in Table 4[8]. We observe that the model trained in the Hindi language exhibits poor performance when applied to Bengali. Conversely, the model trained in Bengali demonstrates subpar results when tested in Hindi. This outcome emphasizes the need to curate new abusive meme datasets for other low-resource languages. By doing so, we can accurately detect multilingual memes in low-resource languages and contribute to the advancement of research in this domain.

## 5.5 Error analysis

In order to understand the workings of the models further, we analyze the inferences of best text-based (XLMR), image-based (VIT), and multimodal (CLIP(L)) classifiers on a small set of examples from the test set. We manually inspect some of the misclassified memes by each classifier and try to provide reasons for these cases.

- *XLMR wins VIT fails*: We observe that when the extracted OCR text contains slurs targeting individuals or communities, or the abusiveness in the text is clear while the image does not carry any meaningful information about the post being abusive, XLMR model

---

[8]Here, for the Bengali meme language, we show the performance of fold 1 with which transferability has been performed.

performs better.

- *VIT wins XLMR fails*: When the meme image represents sexual stereotyping toward women by objectifying their bodies, and the extracted OCR text looks benign, the vision model performs better.

- *CLIP(L) wins, unimodal models fail*: These are the cases when the fusion-based multimodal model predicts the actual label correctly; even the best unimodal text-based and image-based model fails. We observe considering only the extracted text and image individually, the meme does not look abusive. But when both the modalities are used jointly to decide the label, the meme can be easily demarcated as abusive. These are the cases where a better representation can be learnt using the two modalities together.

- *All three models win*: When the extracted OCR text itself is abusive and the image depicts sexual objectification or clear hatred, all the models are successful.

- *All three models fail*: These are the cases when the memes are implicitly abusive and the models miss to capture the context due to the absence of explicit hateful content. Detecting these instances requires understanding sarcasm and complicated reasoning skills or background knowledge about certain situations/events.

We illustrate these findings in Figure 13 by presenting examples of misclassified cases encompassing various scenarios. In Figure 13(a), we encounter a scenario where the vision-based VIT model failed to capture the true message of the meme. The image seemingly portrays the Prime Minister of India, *Narendra Modi*, with a waterfall behind him, giving an impression of benign content; however, the textual component contains a derogatory term, 'Malaun,' directed toward *Narendra Modi*. Based on this textual context, the text-based model successfully classifies the meme as abusive. Proceeding to Figure 13(b), an interesting case emerged where the XLMR model mispredicted the meme. The text lacked any abusive elements, yet the image overtly displayed sexual vulgarity toward women, and the VIT model labeled this as abusive. Figure 13(c) introduces another intriguing scenario. Analyzing the image and text in isolation would not suggest abusiveness. However, when considering both elements, the CLIP

model accurately classifies the meme as abusive and directed toward obese women. Figure 13(d) depicts a meme where the text and image conveys offensiveness toward Pakistanis with the text containing the hateful term 'Pakistan murdabad'. The accompanying image displayed individuals placing their feet on the Pakistani flag, a deeply disrespectful act. Consequently, all models correctly classify this as abusive. Continuing to Figure 13(e), an image portrays a man seemingly experiencing a hangover. However, the meme insinuates that the former cricket coach, *Ravi Shastri*, is obsessed with an alcohol problem and is meant to bully *Ravi Shastri*. Interpreting such memes needs intricate reasoning abilities and contextual knowledge.

## 6 Discussion

The abusive meme detection model can be very effective in content moderation, especially when there is a surge in multimedia content over social media platforms. The auto-discard technique has become increasingly valuable, where hateful content is automatically discarded from posting once flagged by an abuse detection model. Other approaches include alerting the user about a flagged post or lowering the visibility of the flagged post. Finally, moderators can be automatically informed to counter the flagged post so that it dilutes or nullifies the harmful effect of the post. In order to have such an effective flagging, some challenges include the continuous availability of training data, computational costs, and the imperative of upholding the model's sustained accuracy over time.

## 7 Conclusion

This paper presents a new benchmark dataset for Bengali abusive meme detection. Our dataset comprises more than 4K memes collected using keyword-based web search on various platforms. We further evaluated the performance of different classifiers ranging from text-based, image-based, and multimodal models. Our experiments show that using both textual and visual modalities together helps to enhance the performance by learning better representation of the memes. This is further demonstrated by our in-depth error analysis.

## Limitations

There are a few limitations of our work. First, we have only used the OCR extracted textual fea-

tures and visual representation by the pre-trained models to classify the memes. We have not considered additional features such as textual and visual entities, which might improve the model performance. Second, sometimes malicious users intentionally add random noise to their posts to deceive the classification model. We have not tested our model's performance on such adversarial attacks. Third, we observed that the model fails to detect memes with implicit characteristics. Detecting such memes can be challenging, often requiring intricate reasoning abilities and contextual knowledge. This aspect can be taken up for immediate future work. Fourth, the Bengali language incorporates at least two very large dialectal variations: (a) standard colloquial Bengali (spoken in West Bengal) and (b) Bangladeshi (spoken in East Bengal (Bangladesh)). The British colonizers partitioned these regions based on their socioeconomic structure and religion-based demography (Das et al., 2023). The hate lexicons and the target communities vary based on these dialectal variations and are an additional challenge while handling the Bengali language. Based on the abusive meme's origin and the dialect in which it is spoken, the latent target can vary. Dialect-based meme annotation and subsequent performance analysis is an important avenue for future research, and we plan to take this up in future.

## Ethics statement

*User privacy*: Our database comprises memes with labeled annotations and does not include personal information about any user.

*Biases*: Any biases noticed in the dataset are unintended, and we have no desire to harm anyone or any group. We believe it can be subjective to determine if a meme is abusive; hence biases in our gold-labeled data or label distribution are inevitable. However, we are confident that the label given to the data is accurate most of the times owing to our high inter-annotator agreement.

*Potential harms of abusive meme detection*: We observed using both modalities we achieved better performance. While these results look promising, these models cannot be deployed directly on a social media platform without rigorous testing. Further study might be needed to check the presence of unintended bias toward specific target communities.

*Intended use* We share our data to encourage more research on detecting abusive memes in a low-resource language such as Bengali. We only release the dataset for research purposes and do not grant a license for commercial use, nor for malicious purposes.

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

## A  Annotation guidelines

### A.1  Abusive meme annotation

We consider a meme as abusive if –

- It is targeted against a person or group based on protected attributes such as race, religion, ethnic origin, sexual orientation, disability, social sub-groups, caste, or gender.

- It uses derogatory or racial slur words within the meme toward a target community.

- It uses disparaging terms with the intent to harm or incite harm.

- It makes use of idiomatic, metaphorical, collocation, or any other indirect means of expression that are harmful or may incite harm and/or

- It expresses violent communications.

### A.2  Vulgar meme annotation

We denote a meme as vulgar if the meme contains explicit sexual content, offensive slurs, or graphic violence. They may target sensitive topics, make fun of individuals or groups, or contain content that is generally considered offensive or against social norms. Memes with mild profanity or suggestive elements should be considered non-vulgar unless they cross the line into explicit or offensive territory.

### A.3  Sarcasm annotation

We consider a meme as sarcastic if –

- The meme's text conveys a different meaning than its literal interpretation,

- It uses ironic or satirical elements that intend to express something contrary to the literal text.

### A.4  Sentiment annotation

- **Positive sentiment** refers to a positive perception, attitude, or opinion toward a particular subject or situation. A meme should be labeled as positive sentiment if it indicates happiness, joy, excitement, admiration, appreciation, or any other positive emotion.
- **Negative sentiment** refers to an expression or tone that indicates a negative perception or attitude towards a particular subject or situation. Negative sentiment in memes may involve sadness, anger, frustration, disappointment, sarcasm, criticism, or any other negative emotions.

| | | | |
|---|---|---|---|
| কাংলাদেশ | চুদানির পো | ফাকিস্তানিরা | মাইগ্যা |
| কাংলাদেশি | চুদি | বাঁড়াচোদা | মাগিবাজ |
| কাংলাদেশিরা | চুদির | বাইনচোদ | মাঙ্গির পুত |
| কাংলু | চুদির পোলা | বানচোদ | মালাউন |
| কুত্তাচোদা | চুদির ভাই | বারো ভাতারি | মালাউনরা |
| কুত্তার বাচ্চা | চোদ | বেশ্যা | মুল্লা |
| থাঙ্কি মাগী | চোদনা | বেশ্যা মাগী | মুল্লারারেণ্ডি |
| থাঙ্কিরপোলা | চোদনা মাগী | বেশ্যাদের | রেণ্ডিদের |
| থানকি | চোদা | বেশ্যারা | রেণ্ডিরা |
| থানকিদের | তোর মাকে চুদি | বোকাচোদা | রেণ্ডিয়া |
| থানকিরা | ধোন | ভাতার চুদা | রেন্ডিয়া |
| থানকি মাগির পোলা | ধোনের বাল | ভেড়াচোদা | রেন্ডিরা |
| গাড় মারা | নটি মাগি | ভোদার বাল | ল্যাওড়া চোদা |
| গুদ | নটি মাগির ঝি | ভোদার বেটা | শুয়োরের বাচ্চা |
| গুদ মারানি | নটি মাগির পোলা | | হারামথোর |
| গুদের বাল | পতিতা | | হারামজাদা |
| চুতমারানি | পতিতারা | | হারামি |
| চুদনা | ফাকিস্তানি | | হারামির বাচ্চা |
| | | | হিজরা |

Table 5: List of words in the abusive lexicon.

- **Neutral sentiment** refers to an expression or tone lacking strong positive or negative emotion. It indicates a state of indifference, objectivity, or neutrality towards a particular subject or situation. Neutral sentiment in memes may involve statements of fact, observations, general information, or content that does not evoke strong emotions.

### A.5   Target-community annotation

- If the meme directly attacks or targets a specific community, identify and annotate the targeted community.
- Focus on communities based on race, ethnicity, religion, gender, sexual orientation, or any other identifiable social group.
- If the meme does not target a specific community or if the target is unclear, label it as "None."

### B   Keywords used to crawl data

Table 5 presents the compilation of words in the abusive lexicon that we used to crawl memes from the Internet. Since most of these words have no counterpart English translation, we could not provide the corresponding glosses. In addition, Table 6 provides detailed information regarding target community-based keywords.

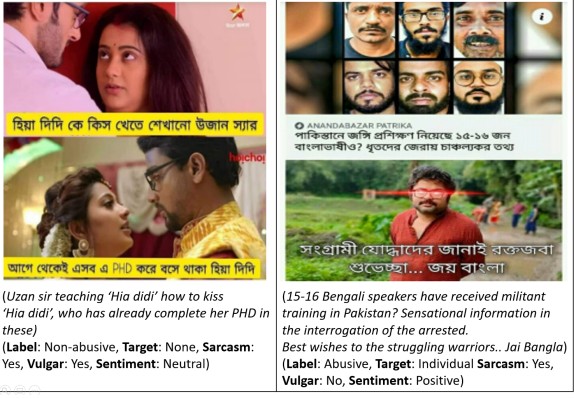

(Uzan sir teaching 'Hia didi' how to kiss 'Hia didi', who has already complete her PHD in these)
(**Label**: Non-abusive, **Target**: None, **Sarcasm**: Yes, **Vulgar**: Yes, **Sentiment**: Neutral)

(15-16 Bengali speakers have received militant training in Pakistan? Sensational information in the interrogation of the arrested.
Best wishes to the struggling warriors.. Jai Bangla)
(**Label**: Abusive, **Target**: Individual **Sarcasm**: Yes, **Vulgar**: No, **Sentiment**: Positive)

Figure 14: Example of abusive & non-abusive memes.

### C   Implementation details

For all the *text-based models* (**m-BERT** (Devlin et al., 2019), **MurIL** (Khanuja et al., 2021), **XLM-Roberta** (Conneau et al., 2020) and **BanglaBERT** (Bhattacharjee et al., 2022)), we extracted pre-trained 768-dimensional feature vectors from the meme text. These vectors were then passed through dense layers for the final classification. Among the *image-based models*, we resized the memes to a size of $224 \times 224 \times 3$, performed Gaussian normalization, and fed them through the pre-trained **VGG16** (Simonyan and Zisserman, 2014) model to obtain 4096-dimensional feature vectors for each meme. Similarly, for the ResNet-152 model, we resized the images and used the pre-trained **ResNet-152** (He et al., 2016) to extract 2048-dimensional feature vectors for each meme.

| Target | Keywords |
|---|---|
| Gender | ছেলে (Boy), ছেলেরা (Boys), ছেলেদের (Boys'), পুরুষ (Male), পুরুষরা (Males), পুরুষদের (Males'), মেয়ে (Girl), মেয়েরা (Girls), মেয়েদের (Girls'), মহিলা (Woman), মহিলারা (Women), মহিলাদের (Women's), ট্রান্স (Transgender), ট্রান্সদের (Transgender), ট্রান্সরা (Transgenders), কিন্নর ( Transgender), কিন্নররা ( Transgenders), কিন্নরদের ( Transgenders') |
| Religion | হিন্দু (Hindu), হিন্দুরা (Hindus), হিন্দুদের (Hindus'), মুসলিম (Muslim), মুসলিমরা (Muslims), মুসলিমদের (Muslims') |
| National Origin | ভারত (India), ভারতের (India's), ভারতীয় (Indian), ভারতীয়রা (Indians), ভারতীয়দের (Indians'), বাংলাদেশ (Bangladesh), বাংলাদেশী (Bangladeshi), বাংলাদেশীরা (Bangladeshis), বাংলাদেশীদের (Bangladeshis'), পাকিস্তান (Pakistan), পাকিস্তানি (Pakistani), পাকিস্তানিরা (Pakistanis), পাকিস্তানিদের (Pakistanis') |
| Individual | খালেদা (Khaleda), মমতা ব্যানার্জি (Mamata Banerjee), হাসিনা (Hasina), নরেন্দ্র মোদী (Narendra Modi), অমিত শাহ (Amit Shah), দেব (Dev), জিত (Jeet), স্রাবন্তি (Srabanti), স্বস্তিকা (Swastika), ভিরাট (Virat), তসলিমা (Taslima), পরী মনি (Pori Moni), মিথিলা (Mithila) |
| Political | আওয়ামী লীগ (Awami League), জাতীয় পার্টি (National Party), বিজেপি (BJP - Bharatiya Janata Party), বিএনপি (BNP - Bangladesh Nationalist Party), তৃনমূল (Trinamool Congress), সিপিএম (CPM - Communist Party of India (Marxist)) |
| Social Sub-groups | ঘটি (Ghoti), ঘটিরা (Ghotis), ঘটিদের (Ghotis'), বাঙাল (Bangal), বাঙালরা (Bangalis), বাঙালদের (Bangalis'), বিহারি (Bihari), বিহারীরা (Biharis), বিহারীদের (Biharis') |

Table 6: Target community-based keywords.

| M | Model | Abusive | | Sentiment | | Sarcasm | | Vulgar | |
|---|---|---|---|---|---|---|---|---|---|
| | | Acc | M-F1 | Acc | M-F1 | Acc | M-F1 | Acc | M-F1 |
| T | m-BERT | 64.21 | 63.12 | 44.57 | 40.31 | 63.07 | 61.90 | 65.71 | 61.68 |
| | MuRIL | 65.61 | 63.97 | **50.16** | 35.63 | 59.61 | 58.95 | 64.35 | 60.25 |
| | BanglaBERT | 64.48 | 62.90 | 46.37 | 42.79 | 61.68 | 60.89 | 63.39 | 59.57 |
| | XLMR | 66.18 | 65.07 | 49.71 | 43.21 | 59.21 | 58.97 | 61.83 | 58.34 |
| I | VGG16 | 67.86 | 65.70 | 47.98 | 40.96 | 63.71 | 62.30 | 72.74 | 67.06 |
| | ResNet-152 | 69.20 | 66.50 | 47.88 | 42.75 | 65.19 | 64.44 | 73.03 | 66.83 |
| | VIT | 69.40 | 67.38 | 45.78 | 42.16 | 66.73 | 66.13 | 73.90 | 69.21 |
| | VAN | 65.22 | 63.67 | 45.63 | 40.63 | 64.35 | 63.86 | 71.25 | 66.21 |
| T + I | MU+RN(C) | 69.08 | 66.32 | 45.93 | 41.53 | 64.97 | 64.20 | 71.87 | 66.49 |
| | XLM+RN(C) | 68.88 | 66.64 | 47.66 | 42.97 | 64.18 | 63.07 | 73.80 | 67.91 |
| | MU+VIT(C) | 68.73 | 67.06 | 47.26 | 42.76 | 65.69 | 65.29 | 72.22 | 68.37 |
| | XLM+VIT(C) | 70.02 | 67.83 | 47.51 | 43.88 | 66.83 | 66.18 | 74.20 | 69.28 |
| | CLIP(C) | **72.81** | **70.51** | 49.02 | **45.59** | **69.05** | **68.28** | **77.49** | **71.66** |
| | MU+RN(L) | 68.51 | 66.02 | 47.61 | 42.84 | 64.16 | 63.64 | 73.23 | 67.25 |
| | XLM+RN(L) | 69.18 | 66.19 | 48.28 | 43.64 | 64.72 | 64.04 | 72.49 | 67.19 |
| | MU+VIT(L) | 67.87 | 66.12 | 46.99 | 42.21 | 65.94 | 65.73 | 71.95 | 67.68 |
| | XLM+VIT(L) | 69.10 | 66.78 | 46.17 | 42.02 | 65.42 | 65.05 | 73.41 | 68.18 |
| | CLIP(L) | 70.39 | 68.45 | 49.24 | 44.39 | 68.58 | 68.27 | 75.09 | 70.29 |

Table 7: Experimental results of different multi-task variants in unimodal and multimodal settings. The best performance in each column is marked in **bold**, and the second best is underlined.

The **VIT** (Dosovitskiy et al., 2021) and **VAN** (Guo et al., 2022) models provided 768-dimensional and 512-dimensional feature vectors respectively for each meme, which were then passed through dense layers for prediction. We extracted 512-dimensional feature vectors for the meme text and image in the multimodal CLIP model. We fed them through two different fusion-based models discussed in section 4. Further details about the other models can be found in section 4.1.

All the models are run for 30 epochs with Adam optimizer, batch_size = 32, learning_rate = $1e-4$. We store the results for the best validation macro-F1 score. All the models are coded in Python using the Pytorch library. We use Ryzen 9, $5^{th}$ gen 12 core processor, a Linux-based system with 64GB RAM and 16GB RTX 3080 GPU.

## D More examples

In Figure 14, we have shown additional examples of abusive memes in contrast to the non-abusive ones.

## E Multi-task classification

Since our dataset has multiple labels for each meme, we aim to understand if we can utilize a multi-task framework for abusive meme detection where vulgarity detection, sentiment analysis, and sarcasm detection act as secondary/auxiliary tasks. To learn $n$ number of tasks simultaneously, after extracting the features from the different models, these are passed through $n = 4$ different linear

layers. Each of these layers are responsible for making predictions for a different type of label. We employ cross-entropy as a loss function to train the network's parameters. As our primary task is to classify memes as abusive or not, we assign loss weight for abusive classification 1, while for the other tasks, we assign a loss weight of 0.5 each.

Table 7 presents the results of all the tasks we got in terms of accuracy and macro-F1 score. Here, we observe the following.

- For the abusive meme detection, CLIP(C) (acc: 72.81, macro-F1: 70.51) achieved the highest performance, and CLIP(L)(acc: 70.39, macro-F1: 68.45) performed the second best.

- Even for other labels also we observe CLIP(C) performs the best and CLIP(L) performs the second best (*sentiment*– CLIP(C): 45.59, CLIP(L): 44.39; *sarcasm*– CLIP(C): 68.28, CLIP(L): 68.27; *vulgar*– CLIP(C): 71.66, CLIP(L): 70.29) in terms of macro F1 score. However, it is important to note that sentiment classification posed a challenge for all models, with their performance falling lower than expected.

- Further, the comparison of these results with those in Table 2 reveals interesting insights. Some models showcase improvement, while others experience marginal performance degradation. Nonetheless, these shifts in performance are not significant. Thus, the decision to pursue a single-task or multi-task approach can be tailored to specific task requirements.