# OpenReview forum: "BanglaAbuseMeme: A Dataset for Bengali Abusive Meme Classification"
_EMNLP/2023/Conference — EMNLP 2023 Main_

### Official Review · Reviewer_zToY · 2023-08-05

**Soundness:** 4

**Excitement:**

4: Strong: This paper deepens the understanding of some phenomenon or lowers the barriers to an existing research direction.

**Paper Topic And Main Contributions:**

The paper addresses the issue of online abuse through the creation and sharing of memes on social media platforms, with a particular focus on Bengali memes. It introduces a new dataset called BanglaAbuseMeme, comprising 4,043 Bengali memes annotated as abusive or non-abusive, with additional labels indicating sentiment, sarcasm, vulgarity, and target community attacked. Moreover, the authors implement and compare several baseline models for automatic detection of abusive Bengali memes, including text-based, image-based, and multimodal models. The results demonstrate that multimodal models, combining textual and visual information, outperform unimodal models. The paper concludes with a qualitative error analysis of misclassified memes by different models, providing insights into their respective strengths and weaknesses.

**Questions For The Authors:**

- Question A: What is the proportion of data that includes implicit abusive memes?
- Question B: It would be helpful if you could provide some examples for the following scenarios: (1) memes that are vulgar but not abusive (Figure 5), (2) memes that have a positive sentiment but are abusive (Figure 6).
- Question C: Table 3 indicates that, except for CLIP, concatenation performs better than late fusion across all methods. Can you provide an explanation for this observation?
- Question D: In the first paragraph of Section 5.2, it is stated that "the performance of the unimodal models varies across all target communities, while the multimodal model consistently reports good performance across all target communities." However, based on Figure 9, it seems that CLIP performs worse on the political target community. Could you clarify this discrepancy?

**Reasons To Accept:**

- The paper introduces a new dataset to facilitate the task of detecting abusive memes in Bengali, a low-resource language. The dataset is planned to be made publicly available for research purposes, offering researchers a valuable resource for studying online abuse in this specific linguistic context. Moreover, the dataset includes additional labels that provide insights into sentiment, sarcasm, vulgarity, and target communities attacked, enhancing the richness of the data for further analysis.
- The authors provide a thorough explanation of the annotation process, offering transparency and insights into how the dataset was labeled. The paper also showcases a comprehensive analysis of the data and its various label sets, providing valuable context and understanding of the characteristics of abusive memes in Bengali.
- The paper conducts a thorough comparison of various baseline models, such as text-based, image-based, and multimodal approaches, for automatic detection of abusive memes.
- The paper is well-written and easy to understand.

**Reasons To Reject:**

- The paper does not clearly demonstrate how the proposed methods compare to previous studies on abusive meme detection, specially on Bengali data.
- The annotation strategy raises concerns, particularly with regards to discussing annotators' errors after every batch. Given the subjective nature of the task, this approach may introduce bias and affect the reliability of the annotations.
- The paper's figures and tables are not well-aligned with the main body, which affects the overall readability. Additionally, the small font size used in tables and figures can make it difficult for readers to comprehend the presented information.

**Reproducibility:**

4: Could mostly reproduce the results, but there may be some variation because of sample variance or minor variations in their interpretation of the protocol or method.

**Reviewer Confidence:**

5: Positive that my evaluation is correct. I read the paper very carefully and I am very familiar with related work.

---

> ### Author Rebuttal · Authors · 2023-08-28
>
> We thank reviewer zToY for providing us with detailed feedback and are really glad that the reviewer found our paper interesting. We are happy to clarify any further concerns.
>
> Q1: How the proposed methods compare to previous studies on abusive meme detection, specially on Bengali data.
>
> Ans: There is no benchmark dataset publicly available on Bengali. Hence we could not perform this exercise. However there are datasets in other languages present performing similar work. We have conducted a comparative analysis that includes our Bengali(bn) language-trained model and a multimodal Hindi(hi) code-mixed harmful/cyberbullying meme dataset (Maity et al.,2022 "A multitask framework for sentiment, emotion and sarcasm aware cyberbullying detection from multi-modal code-mixed memes."), as per your recommendation. Our experimentation focused exclusively on the CLIP(L) model, which exhibited superior performance among all the models considered.
> In this experimental setup, we trained the model using the Bengali language and evaluated its performance using the Hindi language, and vice versa. The subsequent performance comparison is outlined below. We observed that the model trained on the Hindi language exhibited poor performance when applied to Bengali, and conversely, the model trained on Bengali demonstrated subpar results when tested with Hindi. This outcome emphasizes the need to curate new abusive meme datasets for other low-resource languages. By doing so, we can accurately detect multilingual memes in low-resource languages and contribute to the advancement of research in this domain.
>
> | Transfer  | Acc   | M-F1  | F1(A) | P(A)  | R(A)  |
> | --------- | ----- | ----- | ----- | ----- | ----- |
> | hi - > hi | 71.64 | 71.46 | 73.77 | 75.2  | 72.4  |
> | hi-> bn   | 52.28 | 52.28 | 51.87 | 41.85 | 68.19 |
> | bn - > bn | 70.08 | 69.28 | 64.3  | 58.44 | 71.47 |
> | bn -> hi  | 50.46 | 45.07 | 27.86 | 70.44 | 17.36 |
>
> Q2: Annotation strategy raises concerns, particularly with regards to discussing annotators' errors after every batch.
>
> Ans: We followed the standard method for the annotation employed by other researchers who have undertaken similar tasks (Saha et al., 2021 "“Short is the Road that Leads from Fear to Hate”: Fear Speech in Indian WhatsApp Groups.", Pramanick et al., 2021 "Detecting harmful memes and their targets."). These procedures help the annotators in gradually becoming proficient in the annotation task over time.
>
> Q3: Paper's figures and tables are not well-aligned with the main body
>
> Ans:  We shall standardize these issues in the final version of the paper.
>
> Question A: What is the proportion of data that includes implicit abusive memes?
>
> This is a very interesting question. We plan to address this in future work.
>
> Question B: It would be helpful if you could provide some examples for the following scenarios: (1) memes that are vulgar but not abusive (Figure 5), (2) memes that have a positive sentiment but are abusive (Figure 6).
>
> Ans: Thank you for the suggestion. We will add these in the final version.
>
> Question C: Table 3 indicates that, except for CLIP, concatenation performs better than late fusion across all methods. Can you provide an explanation for this observation?
>
> Ans: We believe that this depends on the combination of the dataset and task. In an earlier study by Das et al., 2022 ( "Transfer Learning for Multilingual Abusive Meme Detection."), it has been shown for one language – concatenation performs better than late fusion and for another language the late fusion performed better. However the performance difference was minimal, similar to the results we have observed here. Even when we performed multitasking with additional labels, we observed CLIP(C) performs better (please see our response reviewer xN2X). Overall it can be said that the model’s performance may vary depending on the dataset or task we are performing. A better way to understand this would be to develop an explainable model; this is an important future research direction.
>
> Question D: In the first paragraph of Section 5.2, it is stated that "the performance of the unimodal models varies across all target communities, while the multimodal model consistently reports good performance across all target communities." However, based on Figure 9, it seems that CLIP performs worse on the political target community. Could you clarify this discrepancy?
>
> Ans: Yes for the political target community, the CLIP model's performance is comparatively subpar when contrasted with other models. This discrepancy might be attributed to the possibility that XLMR has been pretrained on data containing a substantial amount of political content, potentially contributing to its enhanced predictive capabilities within this category. A prior study conducted by Nicholls et al. in 2022 ("Better political text classification using large language models.") also supports the notion that XLMR outperforms in political text classification tasks. However, further research would be needed to establish this correlation.

---

### Official Review · Reviewer_UVRc · 2023-08-05

**Soundness:** 4

**Excitement:**

4: Strong: This paper deepens the understanding of some phenomenon or lowers the barriers to an existing research direction.

**Paper Topic And Main Contributions:**

This paper focuses on detecting abusive memes, which combine images and text to threaten or target individuals or communities online, with a specific emphasis on the Bengali language. The authors introduce the "BanglaAbuseMeme" dataset containing annotations for 4,043 Bengali memes, enabling the study of meme dynamics. They implement and compare baseline models for automatic abusive meme detection, including text-based, image-based, and multimodal approaches. The results demonstrate the superiority of multimodal models, with their best-performing model achieving a macro F1-score of 70.51. Through qualitative error analysis, the paper highlights model strengths and weaknesses in handling abusive memes, offering insights into the challenges posed by the Bengali language and cultural context. This work contributes to addressing online abuse and content moderation challenges, particularly in low-resource languages like Bengali.

**Reasons To Accept:**

Introducing a new benchmark dataset for abusive meme detection in Bengali is a valuable and timely contribution, addressing a critical gap in research for low-resource languages. The paper systematically investigates the challenge of identifying harmful content on social media, particularly in the Bengali language, by developing and releasing the "BanglaAbuseMeme" dataset. By implementing and comparing various models, including text-based, image-based, and multimodal approaches, the paper demonstrates significant advancements in abusive meme detection, achieving a high macro F1-score. The qualitative error analysis further enriches the understanding of model performance and limitations. This research not only enhances our ability to curb online abuse but also paves the way for future studies in content moderation and NLP for underrepresented languages.

**Reasons To Reject:**

While the introduction of a new benchmark dataset for abusive meme detection in Bengali holds promise, the paper's contributions might be limited by its focus on a single task within the broader context of content moderation. Additionally, although the multimodal model showcases improved performance, the paper could benefit from a deeper exploration of potential biases and challenges specific to the Bengali language and cultural nuances. The lack of comparative analysis with existing benchmarks for other languages might limit the broader impact of the study. Furthermore, the paper could expand its scope by considering real-world deployment challenges and proposing strategies for addressing potential adversarial attacks or evolving abusive content creation techniques.

**Reproducibility:**

4: Could mostly reproduce the results, but there may be some variation because of sample variance or minor variations in their interpretation of the protocol or method.

**Reviewer Confidence:**

4: Quite sure. I tried to check the important points carefully. It's unlikely, though conceivable, that I missed something that should affect my ratings.

---

> ### Author Rebuttal · Authors · 2023-08-28
>
> We thank reviewer UVRc for providing us with insightful feedback and are glad that the reviewer found our paper interesting. We are happy to clarify any further concerns.
>
> Q1: Deeper exploration of potential biases and challenges specific to the Bengali language and cultural nuances.
>
> Ans: It is undoubtedly an exciting area to work in. The data incorporates at least two very large dialectal variations: (a) standard colloquial Bengali (spoken in West Bengal) and (b) Bangladeshi (spoken in East Bengal (Bangladesh)). The British colonizers partitioned these regions based on their socioeconomic structure and religion-based demography. The hate lexicons and the target communities vary based on these dialectal variations and are an additional challenge while handling the Bengali language. Based on the abusive meme's origin and the dialect in which it is spoken, the latent target can vary. Dialect-based meme annotation and subsequent performance analysis is an immediate future work. We shall add this discussion in the conclusion/limitation section of the paper.
>
> Q2: Comparative analysis with existing benchmarks for other languages
>
> Ans: Although there has been limited research on multilingual abusive meme datasets, we have taken your suggestions into account. We have conducted a comparative analysis that includes our Bengali(bn) language-trained model and a multimodal Hindi(hi) code-mixed harmful/cyberbullying meme dataset (Maity, Krishanu, et al. "A multitask framework for sentiment, emotion and sarcasm aware cyberbullying detection from multi-modal code-mixed memes."), as per your recommendation. Our experimentation focused exclusively on the CLIP(L) model, which exhibited superior performance among all the models considered.
> In this experimental setup, we trained the model using the Bengali language and evaluated its performance using the Hindi language and vice versa. The subsequent performance comparison is outlined below. We observed that the model trained on the Hindi language exhibited poor performance when applied to Bengali, and conversely, the model trained on Bengali demonstrated subpar results when tested with Hindi. This outcome emphasizes the need to curate new abusive meme datasets for other low-resource languages. By doing so, we can accurately detect multilingual memes in low-resource languages and contribute to the advancement of research in this domain.
>
> | Transfer  | Acc   | M-F1  | F1(A) | P(A)  | R(A)  |
> | --------- | ----- | ----- | ----- | ----- | ----- |
> | hi - > hi | 71.64 | 71.46 | 73.77 | 75.2  | 72.4  |
> | hi-> bn   | 52.28 | 52.28 | 51.87 | 41.85 | 68.19 |
> | bn - > bn | 70.08 | 69.28 | 64.3  | 58.44 | 71.47 |
> | bn -> hi  | 50.46 | 45.07 | 27.86 | 70.44 | 17.36 |
>
> Q3: The paper could expand its scope by considering real-world deployment challenges and proposing strategies for addressing potential adversarial attacks.
>
> Ans: Thank you for bringing up this matter. Indeed the abusive meme detection model can be very effective in content moderation especially when there is a surge in multimedia content over the social media platforms. The auto-discard technique, where a hateful content is automatically discarded from posting once flagged by an abuse detection model has become increasingly useful. Other approaches include alerting the user about a flagged post or lowering the visibility of the flagged post. Finally, moderators can be automatically informed to counter the flagged post so that it dilutes or nullifies the harmful effect of the post. In order to have such an effective flagging some of the challenges include continuous availability of training data, computational costs, and the imperative of upholding the model's sustained accuracy over time. We shall add this discussion in the revised version of the paper.

---

### Official Review · Reviewer_xN2X · 2023-08-12

**Soundness:** 3

**Excitement:**

2: Mediocre: This paper makes marginal contributions (vs non-contemporaneous work), so I would rather not see it in the conference.

**Paper Topic And Main Contributions:**

The paper presents a Bengali meme dataset which consists of 4,043 Bengali memes.
They then performed benchmarking of this dataset for  hate classification using basic Multimodal techniques.

**Questions For The Authors:**

Why did not authors perform multitasking as they have labels for  Sarcasm, Vulgar and Sentiment

**Reasons To Accept:**

The paper presents a good resource for multilingual multimodal toxic detection

**Reasons To Reject:**

1. The paper is a resource paper and does not delve into any specific techniques to solve the task.
2. Limited Novelty and contribution apart from the dataset.
3. In section 5.4, the authors mention observations and fail to give proper reasoning.

**Reproducibility:**

5: Could easily reproduce the results.

**Reviewer Confidence:**

5: Positive that my evaluation is correct. I read the paper very carefully and I am very familiar with related work.

---

> ### Author Rebuttal · Authors · 2023-08-28
>
> We thank reviewer xN2X for providing us with insightful feedback. We are happy to clarify any further concerns.
>
> Q1: Multi-tasking results
>
> Ans: Indeed, this was submitted to the "Resources and Evaluation" track since we felt that it was an important resource contribution to the abusive speech research community. However, as per the suggestion, we also performed multi-tasking and below is the table of the results we got in terms of Accuracy and Macro-F1 score for different labels. Here, we observe that for the abusive meme detection, CLIP(C) (Acc: 72.81, M-F1: 70.51) achieved the highest performance, and CLIP(L)(Acc: 70.39, M-F1: 68.45) performed the second-highest. Even for other labels also we observe CLIP(C) performs the best and CLIP(L) performs the second best (Sentiment: CLIP(C): 45.59, CLIP(L): 44.39 | Sarcasm: CLIP(C): 68.28, CLIP(L): 68.27 | Vulgar: CLIP(C): 71.66, CLIP(L): 70.29) in terms of Macro-F1 score. However, it is important to note that sentiment classification posed a challenge for all models, with their performance falling lower than expected. Further analysis, involving a comparison with the results of the Single Task Abusive Meme Detection (Table 2 in the manuscript), revealed interesting insights. Some models showcased improvement, while others experienced marginal performance degradation. Nonetheless, these shifts in performance were not significant. Thus, the decision to pursue a single-task or multi-task approach can be tailored according to specific project requirements. We shall add these new results in the revised version of the manuscript.
>
> | Model      | Abusive |       | Sentiment |       | Sarcasm |       | Vulgar |       |
> | ---------- | ------- | ----- | --------- | ----- | ------- | ----- | ------ | ----- |
> |            | Acc     | M-F1  | Acc       | M-F1  | Acc     | M-F1  | Acc    | M-F1  |
> | m-BERT     | 64.21   | 63.12 | 44.57     | 40.31 | 63.07   | 61.9  | 65.71  | 61.68 |
> | MuRIL      | 65.61   | 63.97 | 50.16     | 35.63 | 59.61   | 58.95 | 64.35  | 60.25 |
> | BanglaBERT | 64.48   | 62.9  | 46.37     | 42.79 | 61.68   | 60.89 | 63.39  | 59.57 |
> | XLMR       | 66.18   | 65.07 | 49.71     | 43.21 | 59.21   | 58.97 | 61.83  | 58.34 |
> | VGG16      | 67.86   | 65.7  | 47.98     | 40.96 | 63.71   | 62.3  | 72.74  | 67.06 |
> | ResNet-152 | 69.2    | 66.5  | 47.88     | 42.75 | 65.19   | 64.44 | 73.03  | 66.83 |
> | VIT        | 69.4    | 67.38 | 45.78     | 42.16 | 66.73   | 66.13 | 73.9   | 69.21 |
> | VAN        | 65.22   | 63.67 | 45.63     | 40.63 | 64.35   | 63.86 | 71.25  | 66.21 |
> | MU+RN(C)   | 69.08   | 66.32 | 45.93     | 41.53 | 64.97   | 64.2  | 71.87  | 66.49 |
> | XLM+RN(C)  | 68.88   | 66.64 | 47.66     | 42.97 | 64.18   | 63.07 | 73.8   | 67.91 |
> | MU+VIT(C)  | 68.73   | 67.06 | 47.26     | 42.76 | 65.69   | 65.29 | 72.22  | 68.37 |
> | XLM+VIT(C) | 70.02   | 67.83 | 47.51     | 43.88 | 66.83   | 6618  | 74.2   | 69.28 |
> | CLIP(C)    | 72.81   | 70.51 | 49.02     | 45.59 | 69.05   | 68.28 | 77.49  | 71.66 |
> | MU+RN(L)   | 68.51   | 66.02 | 47.61     | 42.84 | 64.16   | 63.64 | 73.23  | 67.25 |
> | XLM+RN(L)  | 69.18   | 66.19 | 48.28     | 43.64 | 64.72   | 64.04 | 72.49  | 67.19 |
> | MU+VIT(L)  | 67.87   | 66.12 | 46.99     | 42.21 | 65.94   | 65.73 | 71.95  | 67.68 |
> | XLM+VIT(L) | 69.1    | 66.78 | 46.17     | 42.02 | 65.42   | 65.05 | 73.41  | 68.18 |
> | CLIP(L)    | 70.39   | 68.45 | 49.24     | 44.39 | 68.58   | 68.27 | 75.09  | 70.29 |
>
> Q2: Error Analysis Reasoning
>
> Ans: In Section 5.4, we conducted a thorough manual inspection of misclassified memes and detailed our observations. We illustrated these findings in Figure 13 by presenting examples of misclassified cases encompassing various scenarios.
> In Figure 13(a), we encountered a scenario where the vision-based VIT model failed to capture the true essence of the meme. While the image seemingly portrays the Prime Minister of India, Narendra Modi, with Waterfall behind him, giving an impression of benign content; however, the textual component contains a derogatory term, 'Malaun,' directed toward  Narendra Modi. Based on this textual context, the text-based model successfully classified the meme as abusive.
> Proceeding to Figure 13(b), an exciting case emerged where the XLMR model mispredicted the meme. The text lacked any abusive elements, yet the image overtly displayed sexual vulgarity toward women, and the VIT model labeled this as abusive.
> Figure 13(c) introduces another intriguing scenario. Analyzing the image and text in isolation would not suggest abusiveness. However, when considering both elements, the CLIP model accurately classified the meme as abusive and directed toward obese women.
> Figure 13(d) depicted a meme where the text and image conveyed offensiveness toward Pakistanis. The text contained the phrase "Pakistan murdabad," which is inherently harmful for dissemination on social media. The accompanying image displayed individuals placing their feet on the Pakistani flag, a deeply disrespectful act. Consequently, all models correctly classified this as abusive.
> Continuing to Figure 13(e), an image portrayed a man seemingly experiencing a hangover. However, the meme insinuated that the former cricket coach, Ravi Shastri, is obsessed with an alcohol problem and is meant to bully Ravi Shastri. Interpreting such memes needs intricate reasoning abilities and contextual knowledge. We intend to incorporate this insightful explanation to strengthen the error analysis section in the final version of the paper.
>
> We sincerely hope that you could re-evaluate this paper with goodwill.

---

### Meta-Review · Area_Chair_uf66 · 2023-09-21

**Recommendation:** 5

**Metareview:**

This study introduces a dataset comprising 4,043 memes labeled as either abusive or non-abusive. Additionally, benchmark results using various models are provided.

Reviewers zToY and UVRc both expressed appreciation for the dataset's contribution. Reviewer xN2X raised concerns about the multitask results, which the authors addressed in their rebuttal. The issue of "comparative analysis with existing benchmarks" raised by Reviewer UVRc was also tackled during the rebuttal.

To enhance the paper, please ensure all feedback from the reviewers is addressed.

---

### Decision · Program_Chairs · 2023-10-07

**Decision:**

Accept-Main

**Comment:**

This study introduces a dataset comprising 4,043 memes labeled as either abusive or non-abusive. Additionally, benchmark results using various models are provided.

Reviewers zToY and UVRc both expressed appreciation for the dataset's contribution. Reviewer xN2X raised concerns about the multitask results, which the authors addressed in their rebuttal. The issue of "comparative analysis with existing benchmarks" raised by Reviewer UVRc was also tackled during the rebuttal.

To enhance the paper, please ensure all feedback from the reviewers is addressed.